# Fabrication of EVOH/PANI Composite Nanofibrous Aerogels for the Removal of Dyes and Heavy Metal Ions

**DOI:** 10.3390/ma16062393

**Published:** 2023-03-16

**Authors:** Junshan Zhu, Hang Lu, Jianan Song

**Affiliations:** 1Sinopec Marketing Jiangsu Company, Nanjing 210003, China; 2Research School of Polymeric Materials, School of Materials Science & Engineering, Jiangsu University, Zhenjiang 212013, China

**Keywords:** dyes, heavy metal ions, composite nanofibrous aerogels, PANI nanorods, adsorption

## Abstract

Water pollution caused by the leakage and discharge of pollutants, such as dyes and heavy metal ions, can cause serious damage to the environment and human health. Therefore, it is important to design and develop adsorbent materials that are efficient and multifunctional for the removal of these pollutants. In this work, poly(vinyl alcohol-co-ethylene) (EVOH)/polyaniline (PANI) composite nanofibrous aerogels (NFAs) were fabricated via solution oxidation and blending. The aerogels were characterized by a scanning electron microscope, Fourier transform infrared spectrometry, a contact angle measuring instrument and a universal testing machine. The influences of the introduction of PANI nanorods on the structural properties of aerogels were investigated, and the adsorption performance of aerogels was also studied. The results showed that the introduction of PANI nanorods filled the fibrous network structure, reduced porosity, increased surface hydrophilicity and improved compressive strength. Furthermore, EVOH/PANI composite NFAs possess good adsorption performances for dyes and heavy metal ions: The adsorption capacities of methyl orange and chromium ions (VI) are 73.22 mg/g and 115.54 mg/g, respectively. Overall, the research suggests that EVOH/PANI NFAs have great potential as efficient and multifunctional adsorbent materials for the removal of pollutants from water.

## 1. Introduction

With rapid industrialization, the discharge of industrial wastewater and domestic sewage has become a serious threat to human health and the environment, especially with the presence of dyes and heavy metal ions [1]. Several strategies have been developed to remove these pollutants from water, such as biodegradation, filtration, chemical precipitation and adsorption [2,3,4]. Among those methods, adsorption has been applied widely due to its good universality and high efficiency [5,6]. Conventional adsorbent materials mainly include inorganic carbon-based materials [7], polymeric membrane materials [8] and particulate materials [9]. However, these materials have several drawbacks, including low removal efficiency, poor reusability, high cost and time-consuming processes [10,11], limiting their further application in industry. Therefore, the design and development of low-cost and high-efficiency adsorbents are essential for water purification and environmental protection.

Aerogels are known as a kind of three-dimensional nanoporous structural solid material with low density, high porosity and a large specific surface area [12,13], making them excellent adsorbents for removing dyes and heavy metal ions. However, traditional particle aerogels are composed of a brittle granule skeleton with poor structural continuity, leading to low structural strength and poor mechanical stability [14]. With these aerogels, irreversible structural collapse could occur during the process of water purification and separation, which makes meeting the requirements for the design of low-cost and high-efficiency adsorbents difficult.

Several attempts have demonstrated that a continuous fibrous structure would improve the mechanical properties and structural properties of aerogels [15,16]. In recent years, nanofibrous aerogels (NFAs) fabricated via the assembly of nanofibers, such as cellulose nanofibers [17], poly(vinyl alcohol-co-ethylene) (EVOH) nanofibers [18] and silicon dioxide (SiO_2_) nanofibers [19], have received significant attention. When NFAs were subjected to outside forces, the nanofibrous and pore structure would undergo elastic deformation to avoid stress concentration. After the removal of outside forces, the fibrous and pore structure could recover to the initial shape [20]. Therefore, the NFAs are able to withstand external forces without any structural damage, such as stretching, bending and compressing, during the manufacture and adsorption process. Moreover, nanofibers exhibit good characteristics of solvent transfer and adsorption, which can improve adsorption efficiency and realize efficient swage purification [21].

Polyaniline (PANI) is an efficient adsorbent due to its easy synthesis, remarkable protonation reversibility, good environmental stability and large amounts of amino and imine nitrogen groups, which have been extensively investigated [3]. However, pure PANI (nanoparticles and nanorods) tends to aggregate, limiting its adsorption efficiency [22]. Loading PANI particles onto substrates has been used to overcome this challenge [23]. Xu et al. [24] loaded PANI particles on EVOH nanofibrous membranes via chemical oxidative polymerization. The obtained PANI/EVOH composite nanofibrous membranes showed high adsorption capacity (93.09 mg/g) for chromium ions (Cr(VI)). Mohammad et al. [25] fabricated composite membranes by coating PANI on electrospun nanofibrous membranes. The composite membranes exhibited good adsorption performance for methylene blue. However, the membrane-based substrates still face the drawbacks of low porosity and specific surface area, which cannot afford high adsorption sites.

Herein, the flexible thermoplastic EVOH nanofibers, fabricated by melt extrusion and phase separation, were used to generate EVOH NFA substrates (high porosity and specific surface area), and PANI nanorods obtained via solution oxidation were well distributed into EVOH NFAs. The morphologies, chemical composition, mechanical properties and surface wettability of resultant EVOH/PANI composite NFAs were investigated. The adsorption performance of EVOH/PANI composite NFAs for dyes and heavy metal ions under different conditions (pH, PANI contents, temperature, ionic strength and adsorption time) was also discussed.

## 2. Experimental Section

### 2.1. Materials

A poly(ethylene-co-poly(vinyl alcohol)) (EVOH, model: ET3803) masterbatch with a density of 1.17 g/cm^3^ (poly(vinyl alcohol) content 62 mol%) was supplied by Nippon Synthetic Chemical Industry Co., Ltd. (Tokyo, Japan). Cellulose acetate butyrate ester (CAB, model: 381-20) was acquired from Eastman Chemical Company. Acetone, aniline, ammonium persulfate (APS), citric acid, acetic acid, tert-butanol, sodium dodecyl benzene sulfonate (SDBS), glutaraldehyde (GA), sodium chloride (NaCl), potassium dichromate, Congo red, methyl orange, acid red 88 and methylene blue were acquired from Sangon Biotech Co., Ltd. (Shanghai, China). All commercially obtained chemical reagents were used as received.

### 2.2. Preparation of PANI Nanorods

First, 0.45 mL of aniline and 0.32 g of citric acid were dissolved in 24 mL of deionized water with constant stirring for 30 min to form solution “A”. Then, 1.09 g of APS was dissolved in 12 mL of deionized water and stirred for 10 min to form solution “B”. Solution “A” and solution “B” were mixed and stirred for 1 min. After that, the obtained mixed solution was placed in a refrigerator (2–8 °C). Finally, the mixed solution was washed alternately with deionized water and ethanol several times and dried in a vacuum drying oven at 60 °C for 4 h to obtain PANI nanorods.

### 2.3. Fabrication of EVOH/PANI Composite NFAs

The EVOH nanofibers were generated by melt extrusion and phase separation, which have been proven to construct nanofibers [26,27]. An amount of 1 g of obtained EVOH nanofibers was dispersed into 100 mL of mixture solvent (deionized water and tert-butanol at a volume ratio of 3:1) by homogenization to form EVOH nanofibrous suspensions. PANI nanorods, SDBS, GA and acetic acid were added into EVOH nanofibrous suspensions to obtain an EVOH/PANI composite nanofibrous suspension. The uncrosslinked EVOH/PANI composite NFAs were prepared by freeze drying the composite suspensions. Finally, the EVOH/PANI composite NFAs were obtained by crosslinking them in a vacuum drying oven at 75 °C for 4 h.

### 2.4. Characterization and Measurements

The porosity and densities of composite NFAs were calculated by dividing their mass by their volume. The morphologies of composite NFAs were observed by employing a scanning electron microscope (SEM, Hitachi U8230, Tokyo, Japan). The chemical composition and bonds of NFMs were detected by employing attenuated total reflection–Fourier transform infrared spectroscopy (ATR-FTIR, Thermo Scientific, Nicolet 8700, Waltham, MA, USA). The surface wettability of composite NFAs was measured by using a contact angle analyzer (Kino SL200B, Boston, MA, USA) with a solution volume of 3 μL. The compression behaviors of composite NFAs were measured by an Instron 5969 (Instron, Norwood, MA, USA) equipped with a 100 N load.

### 2.5. Adsorption Performance Measurements

#### 2.5.1. Dyes Adsorption Performance Measurements

The dye solutions (Congo red, methyl orange, acid red 88 and methylene blue) with various concentrations were fabricated by diluting 2000 mg/L of dye solution at 150 r/min. The molecular structure of dyes is shown in Appendix A. About 10 mg of EVOH/PANI composite NFAs was immersed into 8 mL of dye in a 25 °C constant temperature water bath for 24 h. After adsorption saturation, the EVOH/PANI composite NFAs were taken out. The concentrations of dye solutions before and after adsorption were detected, and the characteristic absorbance intensity of dyes (498 nm Congo red, 462 nm methyl orange, 506 nm acid red 88 and 664 nm methylene blue) solution was detected by using an ultraviolet-visible (UV-vis, Perkin Elmer Lambda 35, Waltham, MA, USA) spectrophotometer. The dyes’ adsorption capacities of EVOH/PANI composite NFAs were calculated by using the following formula:(1)$$q=\frac{(C_{0}-C)V}{m}$$
where q is the adsorption capacity (mg/g), $C_{0}$ and C are the concentrations of dye solutions before and after adsorption (mg/L), respectively, V is the volume of dye solutions (mL), and m is the weight of EVOH/PANI composite NFAs (g).

#### 2.5.2. Heavy Metal Ion’s (Cr(VI)) Adsorption Performance Measurements

The Cr(VI) was selected as a model to evaluate the heavy metal ions’ adsorption performance of EVOH/PANI composite NFAs. The Cr(VI) solutions with various concentrations were fabricated by diluting 2000 mg/L of dye solution at 150 r/min. About 10 mg of EVOH/PANI composite NFAs was immersed into a 10 mL solution with various Cr(VI) contents in a 25 °C constant temperature water bath for 24 h. The concentrations of Cr(VI) before and after adsorption were detected at an absorbance intensity of 267.7 nm with respect to the Cr(VI) solution by using inductively coupled plasma mass spectrometry (ICP-AES, Thermo 7400, Waltham, MA, USA). The Cr(VI) adsorption capacity of EVOH/PANI composite NFAs was calculated by using the following formula:(2)$$\eta =\frac{(C_{0}-C)V}{m}$$
where η is the adsorption capacity (mg/g), $C_{0}$ and C are the concentrations of Cr(VI) solutions before and after adsorption (mg/L), respectively, V is the volume of Cr(VI) solutions (mL), and m is the weight of EVOH/PANI composite NFAs (mg).

### 2.6. The Influence Factors on Adsorption Performance

The pH values were adjusted by 0.1 M HCl solution or 0.1 M NaOH solution. The pH value of dyes or Cr(VI) solutions ranged from around 2 to 6. EVOH/PANI composite NFAs with various PANI contents were used to conduct adsorption tests. The temperature was controlled by the temperature control device. The ionic strength was adjusted by adding the NaCl solution with various concentrations. The influences of pH values, PANI contents, temperature and ionic strength on the dye or the Cr(VI) adsorption capacity of EVOH/PANI composite NFAs were explored.

### 2.7. Recyclable Adsorption Performance Measurements

The recyclability experiments were conducted by using NaOH solutions followed by treatment with HCl solutions. The adsorbed EVOH/PANI composite NFAs were placed into 50 mL of the 0.4 M NaOH solution for 30 min and then washed with deionized water. After that, the EVOH/PANI composite NFAs were treated with 0.1 M HCl solution for 30 min. Finally, obtained EVOH/PANI composite NFAs were used for second dyes or Cr(VI) adsorption. The above procedure was repeated 5 times.

## 3. Results and Discussion

### 3.1. Design and Construction of EVOH/PANI Composite NFAs

In this work, high-performance poly(vinyl alcohol-co-ethylene) (EVOH)/polyaniline (PANI) composite nanofibrous aerogels (NFAs) were designed based on the following three criteria: (1) The building blocks (EVOH nanofibers and PANI nanorods) should be well distributed into aerogels; (2) the aerogels should possess good mechanical properties to avoid structural collapse during adsorption applications; (3) the aerogels should exhibit excellent adsorption performance for dyes and heavy metal ions. The first criterion was satisfied by taking deionized water and tert-butanol as dispersing media and using sodium dodecyl benzene sulfonate (SDBS) as a surfactant. To satisfy the second criterion, glutaraldehyde (GA) was selected as a crosslinking agent to create strong bonds between EVOH nanofibers and to ensure that the resulting aerogels had good mechanical properties [28,29]. The third criterion was satisfied by introducing PANI nanorods with good adsorption performance to dyes and heavy metal ions.

As illustrated in Figure 1, the fabrication process of EVOH/PANI composite NFAs included three main steps: the dispersion of EVOH nanofibers and PANI nanorods, freeze drying and heating. The EVOH nanofibers and PANI nanorods were first homogenized in a mixed solution to form uniform composite suspensions. The obtained suspensions were then frozen in a cryogenic refrigerator and dried into uncrosslinked EVOH/PANI composite NFAs with good formability. Finally, EVOH/PANI composite NFAs were generated by heat treatment. The sample names of composite NFAs and the corresponding addition amounts of EVOH and PANI are shown in Table 1.

### 3.2. Morphologies and Structure

The morphology of EVOH/PANI composite NFAs was investigated by SEM analysis. It is obviously observed from Figure 2a that the prepared PANI exists in the form of nanorods with uniform diameter distributions, which demonstrates the successful synthesis of PANI nanorods. As shown in Figure 2b–f, EVOH nanofibers with diameters of 200–500 nm were randomly and isotropically distributed in composite NFAs. The PANI nanorods were distributed uniformly in the pores of EVOH/PANI composite NFAs (Figure 2c–f), indicating the successful introduction of PANI nanorods. With increasing PANI nanorod contents, more PANI nanorods were observed in composite NFAs. The photographs of PANI nanorods and EVOH/PANI composite NFAs are displayed in Appendix A. The color of EVOH/PANI composite NFAs changed from white to green and then to atrovirens. The physical properties of EVOH/PANI composite NFAs are shown in Table 2. The PANI nanorods filled the pores of aerogels, leading to a decrease in porosity.

The ATR-FTIR spectra analysis was taken to characterize the surface chemical compositions of EVOH/PANI composite NFAs. The ATR-FTIR spectra of EVOH NFAs and EVOH/PANI composite NFAs are presented in Figure 3a. The characteristic adsorption peak at 1718 cm^−1^ (corresponding to C=O) was observed in the spectra of EVOH NFAs, which demonstrated a crosslinking reaction between EVOH and GA. Two new adsorption peaks located at 1494 cm^−1^ and 1570 cm^−1^ in spectra of EVOH/PANI composite NFAs were attributed to N-Q=N and N-B-N in PANI nanorods (Q and B represent quinone and benzene rings), respectively [30]. The adsorption peaks of PANI nanorods were found in the spectra of EVOH/PANI composite NFAs, which proved the successful introduction of PANI nanorods in composite NFAs. With increasing PANI nanorod contents in composite NFAs, the intensity of characteristic adsorption peaks of PANI nanorods increased gradually (Figure 3b). PANI nanorods were well dispersed in composite NFAs, which was proven by this phenomenon.

### 3.3. Mechanical Properties

The mechanical properties of EVOH/PANI composite NFAs were investigated by testing their compression. Figure 4a presents the stress–strain curves (ε = 60%) of composite NFAs with various PANI contents. Two obvious deformation regions were found in the curves: a linear elastic behavior region at low strains; a densification region at large strains [31]. With increasing PANI contents, the maximum stress of composite NFAs increased from 3.80 kPa to 9.84 kPa, indicating that the introduction of PANI contents is beneficial to improving the stiffness of composite NFAs.

The EVOH/PANI composite NFAs also exhibited remarkable compression cycling performance. The EP-1 was used to perform cyclic compression tests (Figure 4b). We expected to observe hysteresis loops and plastic deformations (17.25%) after cyclic compression, which are commonly found in highly deformable materials. After 50 cycles, EP-1 could maintain over 92% of the initial maximum stress, which was ascribed to energy dissipation [32]. Moreover, the EVOH/PANI composite NFAs could bear large compressive strain (80%) and completely recover to their original shape upon the release of the loading (Appendix A).

### 3.4. Surface Wettability

The adsorbents were designed for the removal of pollutants from water, so the surface wettability of composite NFAs is a crucial factor. The surface wettability of EVOH/PANI composite NFAs was characterized by the water contact angle. Generally, the water contact angle was <90°, indicating the hydrophilicity of the surface. In contrast, when the water contact angle was <90°, this represented the hydrophobicity of the surface. As shown in Figure 5, the water contact angle of pure EVOH NFAs is 122.0 ± 2.1°, showing good surface hydrophobicity. This is due to the micro- and nano-pore structure on the surface of aerogels that would reduce the water–solid contact area and because air was trapped under the water droplet [33]. After the addition of a small amount of PANI nanorods, the water contact angle decreased from 122.0 ± 2.1° to 0° immediately. With a continued increase in PANI contents, the water contact angle maintained 0° constantly. This is mainly because PANI nanorods can form hydrogen bonds with water and absorb the water droplets on the surface of aerogels [30]. Therefore, EVOH/PANI composite NFAs with good hydrophilicity can be used to absorb dyes and heavy metal ions in water.

### 3.5. Adsorption Performance

#### 3.5.1. Dyes Adsorption

The effect of PANI nanorod contents in EVOH/PANI composite NFAs on the adsorption capacity for dyes was investigated. The adsorption capacity of the EVOH/PANI composite with various PANI nanorod contents for Congo red is shown in Figure 6a. EP-0 almost did not display any adsorption capacity (6.39 mg/g) for Congo red. After the introduction of PANI nanorods, the adsorption capacity increased to 53.83 mg/g. The amino groups of PANI nanorods in composite NFAs easily bind with hydrogen ions in water and become positively charged; they can then absorb the negatively charged Congo red via electrostatic interactions [24]. Pure EVOH did not show any charge in water. The adsorption capacity increased from 53.83 to 67.05 mg/g, when the PANI contents reached 0.15 g. With a continued increase in PANI contents, the adsorption capacity increased very slowly. Therefore, EP-3 (PANI contents is 0.15 g) was taken to conduct further dye adsorption performance tests. As shown in Figure 6b, the effect of dye types on adsorption performance was studied. EP-3 exhibited good adsorption performance for all dyes. Among them, EP-3 exhibited the highest adsorption capacity to methyl orange (73.22 mg/g), followed by Congo red (67.05 mg/g) and acid red 88 (58.75 mg/g). These three dyes are negatively charged in water, and they can be absorbed by EP-3 via electrostatic interactions. Although methylene blue is a positively charged dye, EP-3 showed adsorption performance relative to methylene blue due to the azo structure in methylene blue being coupled with PANI [24]. The mechanism of dye adsorption is shown in Appendix A. The adsorption capacity of EP-3 for dyes is much higher than that of previous reports [34,35,36]. The pH value has little influence on the adsorption performance of dyes (Figure 6c). As displayed in Figure 6d, the adsorption capacity is almost constant, with increasing NaCl concentrations in methyl orange. This is because the small volume of Na^+^ and Cl^−^ cannot break the strong binding forces between methyl orange and EP-3. The influence of methyl orange temperature is also negligible (Appendix A).

The adsorption capacity of methyl orange with various concentrations by EP-3 was investigated, and the results are presented in Figure 7a. In the first 25 min, the adsorption capacity increased quickly with time. Subsequently, the adsorption rate became gentle and reached equilibrium within 50 min, which was faster than most reported adsorbents [37,38,39]. The equilibrium adsorption capacity for methyl orange was 9.88 mg/g, 14.40 mg/g, 46.32 mg/g and 73.22 mg/g for initial solution concentrations of 10 mg/L, 20 mg/L, 60 mg/L and 100 mg/L, respectively. The recyclability of EVOH/PANI composite NFAs is an important factor for practical applications. The recyclability experiments were conducted by using the NaOH solution followed by a treatment with the HCl solution. The desorbed EP-3 was repeatedly placed in methyl orange solutions. As shown in Figure 7b, even after five cycles, the adsorption capacity remained at 91.8% of the initial adsorption capacity, indicating the excellent recyclability of EVOH/PANI composite NFAs.

#### 3.5.2. Heavy Metal Ions Adsorption

The Cr(VI) was chosen as a heavy metal ion model to characterize the heavy metal ions’ adsorption performance. As reported in the literature [40,41], the adsorption of heavy metal ions in water is highly dependent on the pH. The EVOH/PANI composite NFAs were immersed in Cr(VI) solutions with various pH values ranging from 1 to 6. As shown in Figure 8a, the adsorption capacity was 46.91 mg/g when the pH value is 1. This is due to the fact that Cr(VI) mainly existed in H_2_CrO_4_, which has difficulty in being absorbed via electrostatic interactions. As the pH increased to 2, the Cr(VI) primarily existed in HCrO_4_^−^ and Cr_2_O_7_^2−^, leading to increased adsorption capacity. The Cr(VI) was absorbed into the surface of composite NFAs via electrostatic bonding. As the pH continued to increase, resulting in competition between the OH^−^ and CrO4^2−^ for the same composite NFAs, adsorption decreased. Therefore, the following tests were performed at a pH of 2. The adsorption capacity of the EVOH/PANI composite with various PANI nanorod contents for Cr(VI) is shown in Figure 8b. The adsorption capacity increased significantly from 5.50 mg/g to 115.54 mg/g upon increasing PANI contents from 0 to 0.15 g. However, as the PANI content increased beyond 0.15 g, the adsorption capacity increased slowly. The adsorption capacity of EP-3 for Cr(VI) is much higher than that of previous reports [42,43,44]; thus, EP-3 was selected for further adsorption characterizations. Figure 8c shows the adsorption capacity for Cr(VI) at different temperatures. The adsorption capacity increased slightly with an increase in temperature, indicating that temperature is not a crucial factor for the adsorption of Cr(VI). As shown in Figure 8d, the adsorption capacity reduced slightly with the introduction of NaCl. These results reveal that the presence of NaCl has almost no influence on adsorption.

The adsorption capacity of EP-3 for Cr(VI) with various concentrations was also studied, and the results are shown in Figure 9a. As the concentration of the Cr(VI) solution increased, the equilibrium time also increased. The adsorption capacity increased quickly at first, and then the adsorption rate slowed down. The equilibrium adsorption capacity of Cr(VI) was 9.75 mg/g, 19.18 mg/g, 58.02 mg/g and 98.55 mg/g for 10, 20, 60 and 100 mg/L of the initial solution, respectively. Several adsorption–desorption cycles were performed to test the recyclability of EVOH/PANI composite NFAs. As shown in Figure 9b, after five cycles, the adsorption capacity decreased slightly, but it could keep over 92.1% of its initial adsorption capacity.

## 4. Conclusions

In this study, blending was employed to synthesize EVOH/PANI composite NFAs by introducing PANI nanorods into EVOH NFAs. The resulting aerogels demonstrated the combined adsorption properties of PANI nanorods and the structural benefits of aerogels. The introduction of PANI nanorods not only improved the surface hydrophilicity and compressive strength of EVOH/PANI composite NFAs but also provided the capability to adsorb dyes and heavy metal ions. Notably, at a mass ratio of 4:3 for PANI nanorods to EVOH nanofibers, the adsorption capacity of methyl orange and chromium ions (VI) was up to 73.22 mg/g and 115.54 mg/g, respectively. EVOH/PANI composite NFAs also exhibited outstanding cyclic adsorption performance.

## Figures and Tables

**Figure 1 materials-16-02393-f001:**
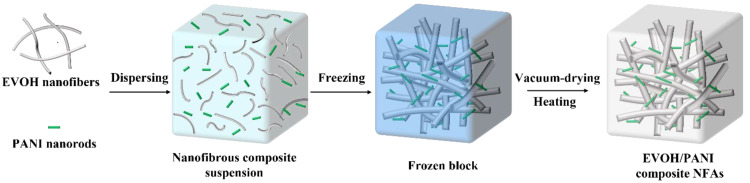
The schematic diagrams of the preparation of EVOH/PANI composite NFAs.

**Figure 2 materials-16-02393-f002:**
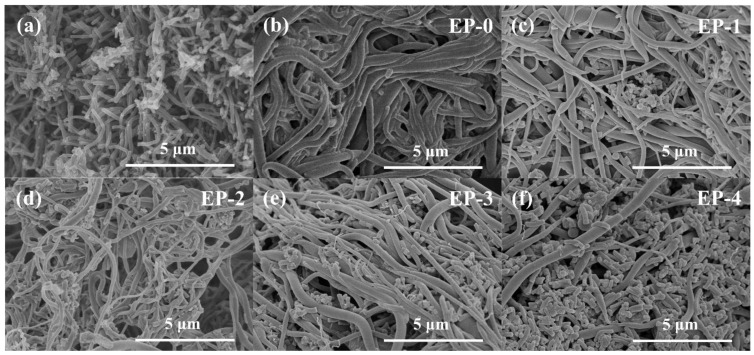
SEM images of PANI nanorods. (**a**) EVOH/PANI composite NFAs; (**b**–**f**) EP-0, EP-1, EP-2, EP-3 and EP-4.

**Figure 3 materials-16-02393-f003:**
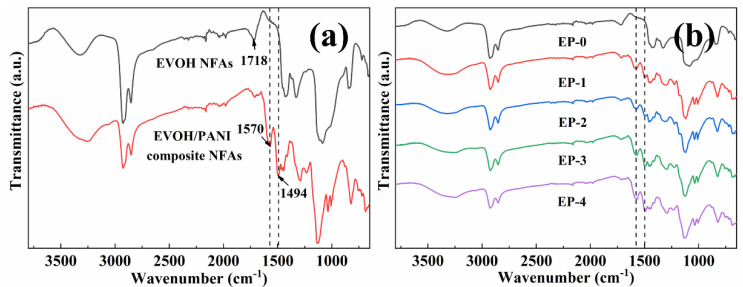
FTIR spectra: (**a**) EVOH NFAs and EVOH/PANI composite NFAs; (**b**) EVOH/PANI composite NFAs with various PANI nanorod contents.

**Figure 4 materials-16-02393-f004:**
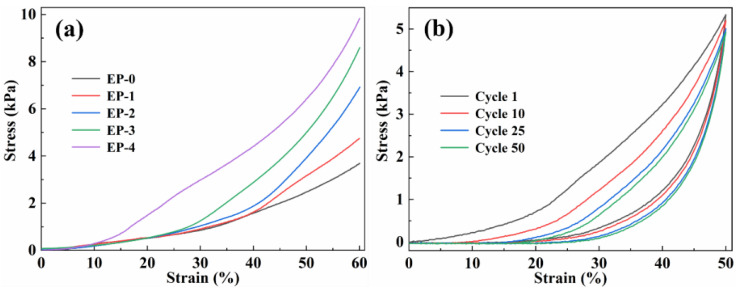
(**a**) Compressive stress–strain curves (ε = 60%) of EVOH/PANI composite NFAs with various PANI nanorod contents; (**b**) cyclic stress−strain curves of EP-1 with ε of 50%.

**Figure 5 materials-16-02393-f005:**
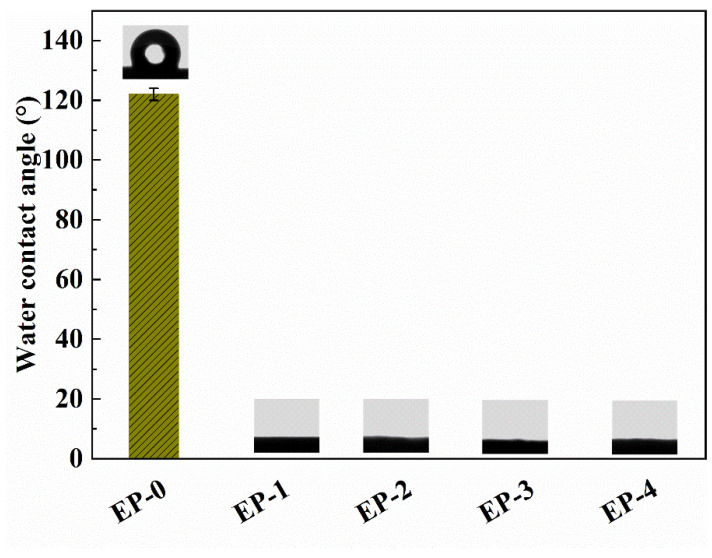
The water contact angles of EVOH/PANI composite NFAs before and after chemical vapor deposition.

**Figure 6 materials-16-02393-f006:**
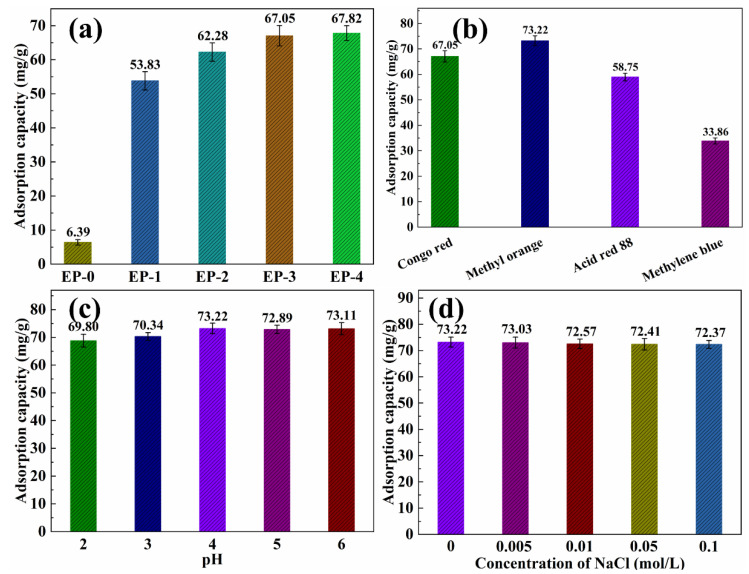
(**a**) Adsorption capacity of EVOH/PANI composite NFAs with various PANI nanorod contents for Congo red; (**b**) adsorption capacity of EP-3 for various dyes; (**c**) adsorption capacity of EP-3 for methyl orange under different pH values; (**d**) adsorption capacity of EP-3 for methyl orange under different ionic strengths.

**Figure 7 materials-16-02393-f007:**
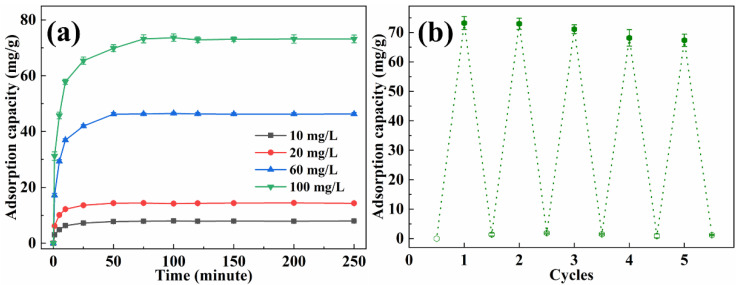
(**a**) The adsorption capacity of EP-3 for methyl orange with various concentrations; (**b**) recyclability test of EP-3.

**Figure 8 materials-16-02393-f008:**
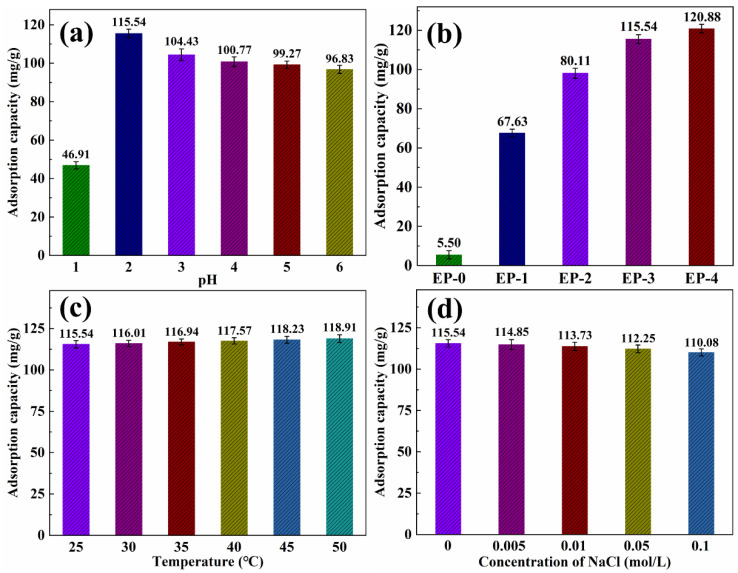
(**a**) Adsorption capacity of EP-3 for Cr(VI) under different pH values; (**b**) adsorption capacity of EVOH/PANI composite NFAs with various PANI nanorod contents for Cr(VI); (**c**) adsorption capacity of EP-3 for Cr(VI) under different temperatures; (**d**) adsorption capacity of EP-3 for Cr(VI) under different ionic strengths.

**Figure 9 materials-16-02393-f009:**
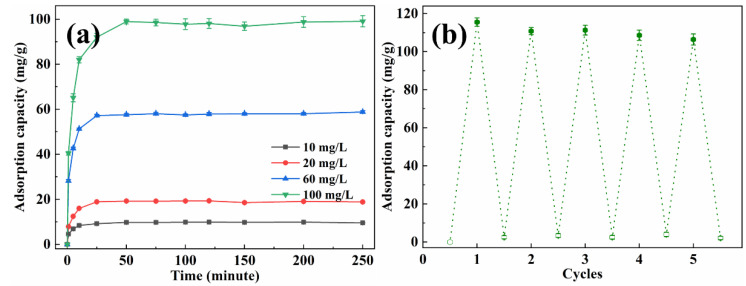
(**a**) The adsorption capacity of EP-3 for Cr(VI) with various concentrations; (**b**) recyclability test of EP-3.

**Table 1 materials-16-02393-t001:** Sample name of EVOH/PANI composite NFAs.

Sample Name	EVOH (g)	PANI (g)
EP-0	0.20	0.00
EP-1	0.20	0.05
EP-2	0.20	0.10
EP-3	0.20	0.15
EP-4	0.20	0.20

**Table 2 materials-16-02393-t002:** The density and porosity of EVOH/PANI composite NFAs.

Sample Name	Density (mg/cm^3^)	Porosity (%)
EP-0	12.12	98.86
EP-1	14.77	98.78
EP-2	19.42	98.42
EP-3	24.03	98.08
EP-4	30.29	97.61

## Data Availability

Data sharing is not applicable.

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
