# Peer review of "Fabrication of EVOH/PANI Composite Nanofibrous Aerogels for the Removal of Dyes and Heavy Metal Ions"

_materials, 2023, doi:10.3390/ma16062393_

Round 1
Reviewer 1 Report
Overall the research is good. Hopefully my advises as below can help to improve the paper quality:
1. Please justify on the concentration of PANI applied in the study.
2. May I know the purpose of showing Figure 2 (a)? What is the difference between this Fig with Fig 2(c) or (d)?
3. Why chose EP-1 (only) to perform cyclic compression test?
4. Title for Fig 6 should be revised to better reflecting the content
5. Any reason Cr(VI) was chosen as a heavy metal ions model to characterize the heavy metal ions absorption performance in this study?
6. Lack of details for the result displayed in Figure 8
- Standardize the format.
- Figure normally should be mentioned before it appear in the text.
Reviewer 2 Report
All my comments are in the attached file.

Round 2
Reviewer 2 Report
The paper was corrected taking into account all my comments. Now the quality of paper was improved and could be published in the presented form.
I have two minor comments:
Subchapter 2.7. should be starts with capital letter.
V = 8mL and m=10 mg for dyes. What about Cr(VI)?? (2.5.2)
Author Response
Dear Materials Editorial Staff:
Thank you for considering our manuscript: “Fabrication of EVOH/PANI Composite Nanofibrous Aerogel for Removal of Dyes and Heavy Metal Ions” that I’m corresponding author.
In the following pages we include responses to the reviewers’ concerns.
Thank you for your continued consideration of our work.
Sincerely,
Jianan Song
Research School of Polymeric Materials, School of Materials Sciences & Engineering
Jiangsu University
songjianan@ujs.edu.cn
Below are the comments from the Reviewers; their comments are in bold text. Our responses are in green text and text added to the manuscript is in red text.
Reviewer: 2
Comments to the Author
The paper was corrected taking into account all my comments. Now the quality of paper was improved and could be published in the presented form.
I have two minor comments:
(1) Subchapter 2.7. should be starts with capital letter.
Thank you for pointing this out. We revised it.
2.7 Recyclable adsorption…
(2) V = 8mL and m=10 mg for dyes. What about Cr(VI)?? (2.5.2)
Thank you for pointing this out. We revised it
About 10 mg EVOH/PANI composite NFAs were immersed into 10 mL solution with…
We tried our best to improve the manuscript and made some changes in the manuscript. These changes will not influence the content and framework of the paper.
We appreciate for Editors/Reviewers’ warm work earnestly, and hope that the correction will meet with approval.
Once again, thank you very much for your comments and suggestions.
